# Employing the Multivariate Edmonton Scale in the Assessment of Frailty Syndrome in Heart Failure

**DOI:** 10.3390/jcm11144022

**Published:** 2022-07-12

**Authors:** Karolina Studzińska, Piotr Wąż, Anna Frankiewicz, Iwona Stopczyńska, Rafał Studnicki, Rita Hansdorfer-Korzon

**Affiliations:** 1Department of Physiotherapy, Faculty of Health Sciences, Medical University of Gdansk, 7 Dębinki Street, 80-211 Gdansk, Poland; rafal.studnicki@gumed.edu.pl (R.S.); rita.hansdorfer-korzon@gumed.edu.pl (R.H.-K.); 2Department of Nuclear Medicine, Medical University of Gdansk, Tuwima 15, 80-210 Gdansk, Poland; piotr.waz@gumed.edu.pl; 3First Department of Cardiology, Medical University of Gdansk, 80-211 Gdansk, Poland; frankiewicz@gumed.edu.pl (A.F.); istopczynska@gmail.com (I.S.)

**Keywords:** frailty, heart failure, Edmonton Frailty Scale

## Abstract

Background: Frailty syndrome (FS) is a syndrome characterized by a reduction in the body’s physiological reserves as a result of the accumulation of reduced efficiency of many organs and systems. Experts of the Heart Failure Association of the European Society of Cardiology (ECS) emphasize the need to assess frailty in all patients with heart failure (HF). There is no specific scale dedicated to this group of patients. The aim of the study was to assess the occurrence of the frailty syndrome in heart failure using the multidimensional Edmonton Frailty Scale (EFS). Methods: The study included 106 patients diagnosed with heart failure with reduced left ventricular ejection fraction (LVEF < 40%). The average age was 62.6 ± 9.7 years. Most of the patients (84%) studied were men. In 70 people (66%), the cause of heart failure was coronary artery disease. The study group included patients admitted to hospital on a scheduled basis and with exacerbation of heart failure. Frailty was measured using the EFS before discharge from the hospital. Demographic, sociodemographic and clinical data were obtained. A 12-month follow-up period was included in the project. The number of readmissions after 6 and 12 months was assessed. Results: A correlation was observed between the New York Heart Association (NYHA) functional class and the occurrence of frailty—this applies to the assessment at the beginning and at the end of hospitalization. When analyzing the age of the patients in relation to frailty, a statistically significant difference was obtained. The youngest group in terms of age were non-frail patients. Hospitalization of people prone to development of the frailty syndrome and diagnosed with the FS was significantly more often associated with the occurrence of complications during hospital stays. Rehospitalizations for exacerbation of heart failure were much more frequent in patients with frailty. Conclusions: Assessment and monitoring of the state of increased sensitivity to the development of frailty or FS in patients with heart failure should influence the differentiation of clinical management. The Edmonton Questionnaire may be a helpful tool for the assessment of frailty in hospitalized patients with HF.

## 1. Introduction

Effective diagnosis and treatment of heart failure (HF) contributes to the survival rate of patients with the advanced form of the disease. The above trend is also associated with an increase in the number of patients with a coexisting frailty syndrome (FS). According to the definition, the FS (also known as weakness or frailty) is a syndrome characterized by a reduction in the body’s physiological reserves as a result of the accumulation of reduced efficiency of many organs and systems [1]. In order to describe the weaknesses, two concepts have been developed which are now quoted in the literature. These are: a phenotypic model—a physical weakness, and a deficits accumulation model—weaknesses in a multidimensional context [2]. According to the authors of the first theory, frailty is diagnosed when at least three of the following five deficits are present: unintentional weight loss, subjective feeling of fatigue (exhaustion), slowing down of the walking speed, weakening of the hand grip, and limitation of physical activity [3]. The concept in a multidimensional approach enables the presentation of the syndrome in a broader perspective, taking into account the relationships between the physical, mental, and social spheres. In this context, weakness means a condition associated with the presence of deficits at least at one level of human functioning [4]. The need to consider frailty in the treatment of patients with HF is emphasized by the authors of international guidelines and recommendations of the ESC on the diagnosis and treatment of acute and chronic heart failure [5,6]. Awareness of the presence of FS and the associated risk of poor prognosis may influence therapeutic decisions and the organization of patient care. In 2019, the Heart Failure Association of the European Society of Cardiology took a position on this issue. They define frailty as “a multidimensional dynamic state, independent of age, which makes a patient with heart failure susceptible to stressors” [7]. The occurrence of a factor (stressor), e.g., infection, drug regime change, is associated with a disproportionate reaction, excessive deterioration of health, and thus increases the risk of decompensation. Complications of the weakness include reduced fitness and independence, greater risk of falls, fractures, greater susceptibility to disease and more frequent hospitalizations, as well as the need for institutional care and premature death [8]. Due to the dynamic nature of the changes, attention was drawn to the need to identify people predisposed to develop weakness with the view to implementing appropriate preventive measures earlier [9,10]. According to the definition proposed by the association, frailty in HF should be diagnosed based on a multidimensional, holistic approach. Assessment of weakness, the basis of physical limitations, is associated with the risk of overlapping shared symptoms such as fatigue, reduced exercise tolerance, and reduction in muscle mass, which may lead to difficulties in the diagnosis of frailty in HF [7]. The definition of the parameter of unintentional weight loss as a component of physical weakness may be an additional problem due to the need to take and periodically modify diuretics. Studies which compared the assessment of frailty in a multivariate context with the phenotypic approach in patients of ≥65 years of age with HF, at 24 months of follow-up, showed that the results of the multivariate scale more accurately predicted adverse effects such as: rehospitalisation, disability, or death during the follow-up period [11]. As suggested by the Heart Failure Association of the European Society of Cardiology, due to the high frequency of occurrence, as well as the impact of frailty on the prognosis, treatment options, and quality of life, the assessment should cover all patients regardless of their chronological age. 

There are many tools in the literature aiding the study of frailty syndrome which incorporate a multi-dimensional approach. However, there is no specific scale dedicated to patients with HF [10,12].

The aim of the study was to assess the occurrence of frailty syndrome in heart failure using the multidimensional Edmonton Frailty Scale (EFS).

## 2. Materials and Methods

### 2.1. Study Participants

This study includes 106 patients with a basic diagnosis of heart failure with reduced left ventricular ejection fraction (LVEF < 40%), hospitalized in the Pomeranian Voivodeship in 2016–2020. The study group included patients admitted to hospital on a scheduled basis (50%) and patients with exacerbation of heart failure (50%).

The exclusion criteria from the study are as follows:patient’s lack of consent to participate in the studyLVEF > 40%other diagnosis significantly burdening the prognosis within one year (e.g., active neoplastic disease, acute coronary syndrome, stroke up to 3 months, pulmonary hypertension)depression diagnosis on the Geriatric depression scale according to Yesavage (GDS) [13].diagnosis of cognitive dysfunction with dementia—assessment based on the Mini-Mental State Examination (MMSE) questionnaire [14].

### 2.2. The Course of the Study

Before discharge from hospital, each patient underwent the following tests and examinations:interview (demographic, sociodemographic, and clinical data were obtained, i.e., New York Heart Association (NYHA) functional class, etiology of the heart failure, presence of comorbidities, reason for planned hospitalization, and the course of hospitalization). The obtained information was verified based on the current medical documentation. Information on the ejection fraction (EF) was obtained from the echocardiographyassessment of severity of the frailty syndrome—Edmonton Frailty Scale questionnaire.evaluation of the occurrence of depression using the Geriatric depression scale according to Yesavageassessment of cognitive functions—MMSE Mini-Mental State Examination questionnaire.Information on hospital readmissions after 6 and 12 months was obtained from telephone interviews conducted with study participants.

### 2.3. Research Tools

The Edmonton Frailty Scale(EFS) was developed by D. Rolfson et al. from the University of Alberta, in Edmonton in 2006, Canada [15]. The original version of the questionnaire includes 11 items: two practical tasks and nine closed questions. The “clock drawing test” was used to assess cognitive impairment on this scale, while the “stand up and walk” test was used to determine balance, mobility, and the risk of falling. The remaining issues were in the form of closed questions and concerned general health, functional independence, social support, medication used, nutrition, mood, and continence. The maximum possible score was 17—this result indicated an advanced state of frailty. Depending on the score in the questionnaire, the following division was distinguished:0–4 points—no frailty5–6 points—particularly sensitive people predisposed to presenting frailty7–8 points—mild frailty9–10 points—moderate frailty11 points or more—serious weakness.

### 2.4. Statistical Analysis

Statistical analysis was performed using the functions and procedures of the R package [16,17,18,19,20]. For quantitative variables, basic statistics were calculated, i.e., mean and standard deviation (when the data set came from a normally distributed population) or the median and the first and third quartiles (for sets for which the Shapiro–Wilk test result was statistically significant). The differences between the two groups of quantitative variables were tested using t-student test or the Wilcoxon test. The kind of the above-mentioned tests (and additional options) were selected depending on the *p*-value of the Shapiro–Wilk test and the homogeneity of variance test. In the event of a larger number of groups, the non-parametric Kruskal–Wallis test was used. In the case of qualitative variables, the basic information that was determined was the frequency of occurrence in the collected research material. Fisher’s Exact test for count data was used to determine if there was a significant difference between the observed and expected frequencies in the designated qualitative variables. One-dimensional and multiple ordinal regression models were created for the variables describing the existence of comorbidities of the studied patients. For each of the above-mentioned tests, the significance level was set at α = 0.05.

## 3. Results

The mean age of 106 patients diagnosed with HF was 62.6 ± 9.7 years. Most of the 89 patients (84%) studied were men. In 70 people (66%), the cause of HF was coronary artery disease. Only 6 patients (5.7%) were not burdened with comorbidities. The studies included arterial hypertension, diabetes, hyperthyroidism or hypothyroidism, chronic kidney disease, stroke, lower limb atherosclerosis, and chronic obstructive pulmonary disease as additional comorbidities. At the beginning of hospitalization, as many as 59 patients (55.6%) were assigned to NYHA class III and III/IV, while at discharge to NYHA class III, 18 patients (17.3%) and 58 patients (55.8%) were assigned to class II. NYHA. Two people died during hospitalization.

Complications during hospitalization concerned 21.7%, i.e., 23 patients. Incidence of infection, bleeding, trauma, and all other complications combined were included.

On the basis of the obtained number of points in the Edmonton scale, three groups of patients were distinguished for the purpose of this study: non-frail, pre-frail, and frail.

Edmonton: 

non-frail (0–4 points) 46 (43.4%)

pre-frail (5–6 points) 47 (44.3%)

frail (≥7 pts) 13 (12.3%)

The general characteristics of the patients are presented in Table 1.

Basing on the qualitative data analysis, contingency tables were created. The columns of these tables correspond to the Edmonton frailty groups (non-frail, pre-frail, frail) and the rows correspond to the groups of such variables as gender, reasons for admission, aetiology of heart failure, NYHA (at the beginning and end of hospitalization), comorbidities, complications. Using Fisher’s Exact Test for Count Data, it was checked whether a relationship between the above-mentioned variables exists. Only the results that are statistically significant are presented (Table 2, Table 3 and Table 4).

More than half of the patients who did not show frailty symptoms at the time of admission to the hospital belonged to class NYHA II (24 people) and NYHA II/III (3 people). Patients diagnosed with FS are mainly NYHA III and NYHA III/IV.

Hospitalization of patients prone to the development of the frailty syndrome or diagnosed with the frailty syndrome was significantly more often associated with the occurrence of complications during the hospital stay. Complications that occurred during hospitalization concerned almost half (46%—six people) of patients with the frailty syndrome, while in the group of patients without the frailty syndrome, complications occurred in only 13% (six people) of patients.

Analysing the data collected at the end of hospitalization for 104 patients, a statistically significant difference was obtained between the Edmonton score and the NYHA e classification (final).

In order to describe the results obtained from the presented contingency tables with higher accuracy, the collected values were used to create an association plot along with the Pearson residuals [21]. In the association plot, each cell of the contingency table is represented by a rectangle. The height of this rectangle is proportional to the Pearson residuals, and the width is proportional to the root of the expected value. The rectangles in each row are presented relative to the baseline, which means no difference between the value from the contingency table and the expected value. The rectangles above the line mean that the value in the contingency table is larger than expected value, and the rectangles below the line mean that the value in the table is smaller than the expected value (Figure 1, Figure 2 and Figure 3).

In another test, the full information contained in the Edmonton ordinal variable was used. Values of the Edmonton ordinal variable were divided into groups with respect to the categorical NYHAb variable and next NYHAe. Using the Kruskal–Wallis test, *p*-value = 0.00091 (NYHAb) and *p*-value < 0.00001(NYHAe) were obtained. with the assumed significance level, indicating there are differences between Edmonton values in individual groups of the NYHA variable. The Kruskal–Wallis test was also used to analyse the age of patients in the Edmonton groups. The median value (first quartile; third quartile) for the age of the studied group of patients, with the division into Edmonton groups, are as follows: non-frail 60.5 (53.25; 65.5), pre-frail 64 (58.5; 70), and frail 63 (57; 78). The test was statistically significant (*p*-value = 0.0323). Moreover, in the post-hoc test, statistical significance was demonstrated between the non-frail and frail groups (*p*-value = 0.04141) and the non-frail and pre-frail groups (*p*-value = 0.02566)—Figure 4. The youngest patients in the non-frail group.

Analogously, statistically significant differences were obtained for qualitative variables such as: comorbidities (*p*-value = 0.03370 Wilcoxon test)—Figure 5, complications (*p*-value = 0.02126 Wilcoxon test).

All patients with heart failure and frailty syndrome 12.3% (13 patients) were diagnosed with additional comorbidities. Of the patients, 98% (46 patients) susceptible to developing the frailty syndrome also had additional diseases. The ordinal regression model was used to determine the relationship between the groups of the Edmonton variable and dichotomous variables describing the existence of comorbidities of the studied patients. Eight models were created in which the dependent variable were groups of the Edmonton variable and the independent variable was one of the eight comorbidities listed in Table 1. The coefficients of the model are statistically significant only for the variable ‘chronic kidney disease’ (*p*-value = 0.00004). The odds ratio OR = 6.6423 was determined using the value of the coefficient for the variable ‘chronic kidney disease’ (1.8935). The obtained result indicates that the odds that a person suffering from chronic kidney disease will be in the group with higher values of the Edmonton variable is over 6.6 times higher than for patients who do not suffer from the above-mentioned disease. The predicted probability of patients suffering from chronic kidney disease belonging to particular groups of the Edmonton variable is presented in Table 5.

A multivariate model was also developed with all comorbidities studied as independent variables. In this model, the determined coefficients were statistically significant (*p*-value = 0.00002 and *p*-value = 0.0412, respectively) for the variables ‘chronic kidney disease’ and ‘hypertension’. The probability that a person suffering from chronic kidney disease will be in the group with higher values of the Edmonton variable is almost eight times more likely, and in the case of hypertension, three times higher than for patients who do not suffer from the above-mentioned diseases.

Again, the Kruskal–Wallis test was used to analyze rehospitalization in each Edmonton´s group. The results were statistically significant after 6 months (*p*-value = 0.02629) and 12 months (*p*-value = 0.03518). A post hoc test showed a statistically significant difference in the number of rehospitalizations between the non-frail and frail groups after 6 (Figure 6) and 12 months (Figure 7), respectively. Patients with frailty syndrome were much more frequently hospitalized at this time. Furthermore, the pre-frail and frail groups were significantly statistically different after 6 (*p*-value = 0.03774) and after 12 months compared to the non-frail and pre-frail groups (*p*-value = 0.04295).

## 4. Discussion

Patients with HF show a 6-fold higher risk of developing the weakness syndrome [7]. The susceptibility and increased sensitivity to the development of the syndrome (the pre-frail group) concerns 46% of patients [22]. The estimated overall frequency of frailty in HF is approximately 45%. However, in studies using tools assessing physical limitations, it is 42.9%, while in those using a multidimensional approach, it is 47.4% [23]. In other studies on HF, the authors describe the frequency of severity of frailty in the range of 15–79% [24]. In the presented study, 44.3% of the respondents showed susceptibility to developing frailty, and 12.3% to frailty. The diversity of the study groups presented in the papers contributes to such a large span of the obtained results. It manifests itself, inter alia, in different clinical conditions of the patients, disease advancement, age, study site, and the use of a variety of tools applied in assessing the frailty syndrome. The described problem influences the limitations of comparing the inter-centre results and comparison of available publications. Nevertheless, it is a foundation for a discussion on the necessity of accounting for the appropriate selection of study groups. Despite the presented difficulties, Zhang and co-authors conducted a systematic review and a meta-analysis which showed that frailty, measured by various scales in heart failure (e.g., the weakness phenotype according to Fried, Frailty Index, Canadian Study of Health and Aging Clinical Frailty Scale), is a significant predictor of all-cause mortality and re-hospitalization [25].

In the other study of HF, the Edmonton scale had already been used to assess frailty [26]. The group in the study consisted of outpatients with a chronic disease (median age was 76), and the diagnosis of frailty in the above study was associated with scoring at least 8 points on the EFS. The maximum number of points on this scale is 17. with a total score of 6–7, the patient was qualified as belonging to the pre-frail group. There are also examples in the literature where the score range on the same scale is different. In the described work, as in several other publications [27,28], scoring more than 7 points was tantamount to the diagnosis of frailty. The pre-frail group consisted of patients within a score range of 5–6. The differences in point thresholds also make it difficult to compare the questionnaire results.

Based on the available literature, it is possible to analyse the group of hospitalized patients with heart failure. Professor Uchmanowicz’s research, covering 330 patients hospitalized for heart failure, showed that there was a positive correlation between the frailty syndrome and the number of patient re-hospitalizations per year [29]. In univariate correlation analysis, treatment with diuretics, higher NYHA class, and lower left ventricular ejection fraction were predictors of more hospitalizations. Patients who were re-hospitalised three or more times showed a higher functional class according to NYHA: IV class in 49.5% of patients vs. 36.5% hospitalized up to twice a year (*p* < 0.001). Moreover, a lower ejection fraction (<40%) was found in the first group [29]. In another study [30], the diagnosis of frailty using the Clinical Frailty Scale (CFS) was associated with a two-fold higher risk of death within one year after hospitalization because of the acute heart failure. The obtained result in this work is consistent with the available research in this area.

When assessing the course of hospitalization, more frequent complications during hospital stay were noticed in patients with FS compared to patients who did not show signs of weakness. According to Kang et al., the presence of frailty may be associated with a poor prognosis during hospitalization of patients more than 60-years old with HFrEF [31]. The weakness compared to the pre-frail group was associated with worse outcomes of inpatient treatment and a significantly higher incidence of adverse events during hospitalization. Moreover, patients with frailty tend to present higher in-hospital mortality rates, and the main cause of these patients’ deaths was multi-organ failure, indeed, it was significantly more frequent in weakened patients vs. pre-frail patients. The conclusions of subsequent studies regarding the discussed issue are as follows: patients with a higher risk of weakness are more prone to in-hospital mortality, with longer and more costly hospital stays. Moreover, there is a greater percentage of these patients discharged to nursing homes [32].

The literature on the subject emphasizes the commonality of additional chronic diseases in patients with HF and FS, which also has a significant impact on the prognosis in this group of patients [33,34]. Similarly, in this study, all patients diagnosed with frailty were additionally burdened by comorbidities. It should be noted, however, that frailty may also apply to those who have not been diagnosed with any chronic disease [3]. Data from the Longitudinal Aging Study Amsterdam (LASA) revealed that community-based elderly people with HF are exposed to an increased risk of weakness, irrespective of other factors such as gender, age, and multiple morbidities (the median in follow-up was 8.4 years) [35].

Analysing the relationship between the NYHA classification and the occurrence of FS, it would seem obvious that the higher NYHA class, i.e., the severity of symptoms and exercise intolerance, is a greater tendency to reveal frailty features. Such results were obtained in our own study. Patients with frailty are NYHA III and III/IV at the beginning of hospitalization and mainly NYHA III at discharge. However, in the previously cited Denfeld meta-analysis covering 26 studies and concerning 6896 patients, no relationship was found between the NYHA functional classification and the FS [23]. Moreover, the relationship between the occurrence of frailty and the age of patients was not revealed in the above-mentioned study. On the other hand, many authors emphasize the importance of age in the presence of frailty in HF [36,37,38,39,40,41]. The frequency of diagnosis of both these problems increases with age and concerns 3.2% of patients aged 65–70 years and over 23% of patients aged 90 and older [39]. There are also studies reporting the occurrence of frailty in younger patients (<60 years old) with heart failure [2,42]. Experts of the Heart Failure Association of the European Society of Cardiology emphasize that frailty cannot be seen as a synonym for aging, physical limitations, and disease severity, and that routine frailty assessment should be part of an overall management plan for all patients with HF. Additionally, the experts indicated that the assessment should be performed with the use of a multidimensional tool which, in addition to physical limitations, takes into account psychological, social, and clinical factors. The obtained results concerning the course of hospitalization and rehospitalization are consistent with the results of other authors. On this basis, we may conclude that the EFS is a useful, simple tool for application in the hospital setting. Due to its multidimensional nature, the information obtained from the study can be used by members of the medical team to determine and organize the care and treatment of patients with heart failure during hospitalization. It can also provide valuable information about the patient to physicians, nurses, physiotherapists, and psychologists.

This study has limitations that need to be highlighted. Among the most important of these is the small size of the study group. Consequently, the number of patients diagnosed with frailty was also limited. The second limitation was that patients from only one centre participated in the project.

The present study also has its strengths. In the study, a multidimensional frailty scale was used to assess FS. Moreover, the study involved a limited group of patients—only those with reduced ejection fraction—and was conducted in hospital conditions. The assessment of frailty in all patients irrespective of age should be considered a major asset.

## 5. Conclusions

Assessment and monitoring of the state of increased sensitivity to the development of frailty or FS in patients with HF should influence the differentiation of clinical management. Our results confirm that patients in the pre-frail group, and the presence of frailty itself, may be associated with higher rates of complications during hospitalization.There is a need to standardize the research in terms of the research tools used in the described group of patients.The Edmonton Questionnaire may be a helpful tool for the assessment of frailty in hospitalized patients with HF. There is a need for further research involving larger groups of study subjects.

## Figures and Tables

**Figure 1 jcm-11-04022-f001:**
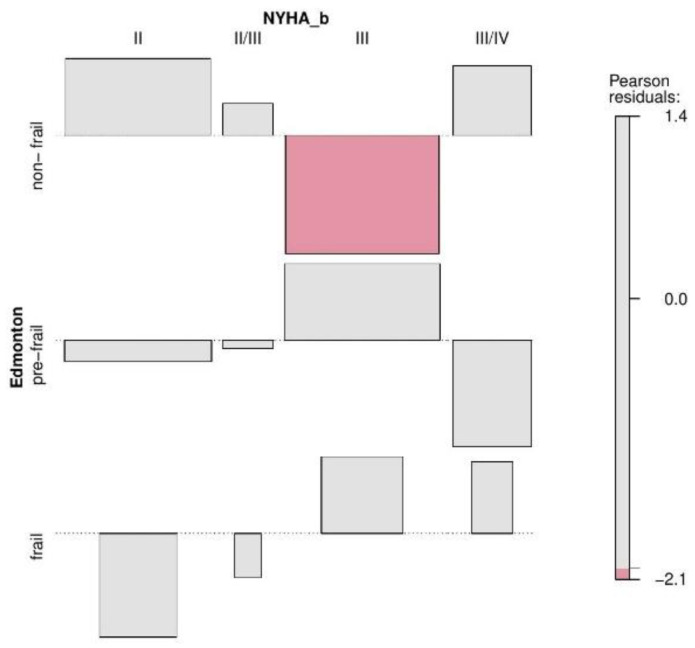
Association plot presenting the multidimensional relationship between NYHA_b and Edmonton subgroups.

**Figure 2 jcm-11-04022-f002:**
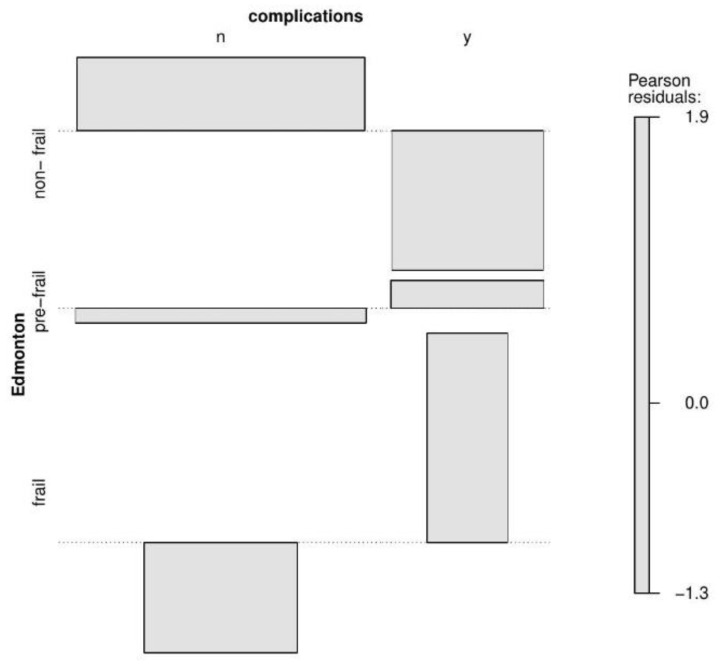
Association plot presenting the multidimensional relationship between Complications and Edmonton subgroups.

**Figure 3 jcm-11-04022-f003:**
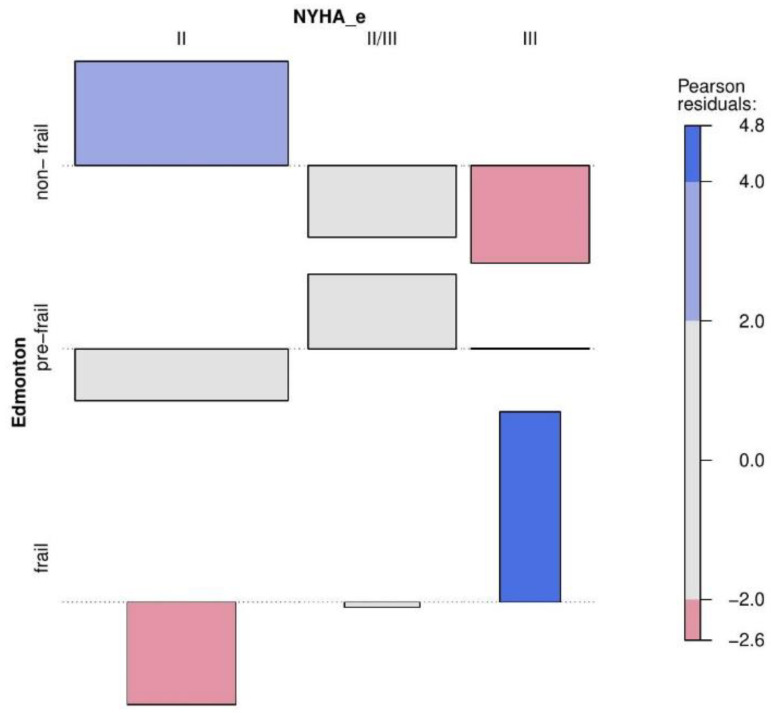
Association plot presenting the multidimensional relationship between NYHA_e and Edmonton subgroups.

**Figure 4 jcm-11-04022-f004:**
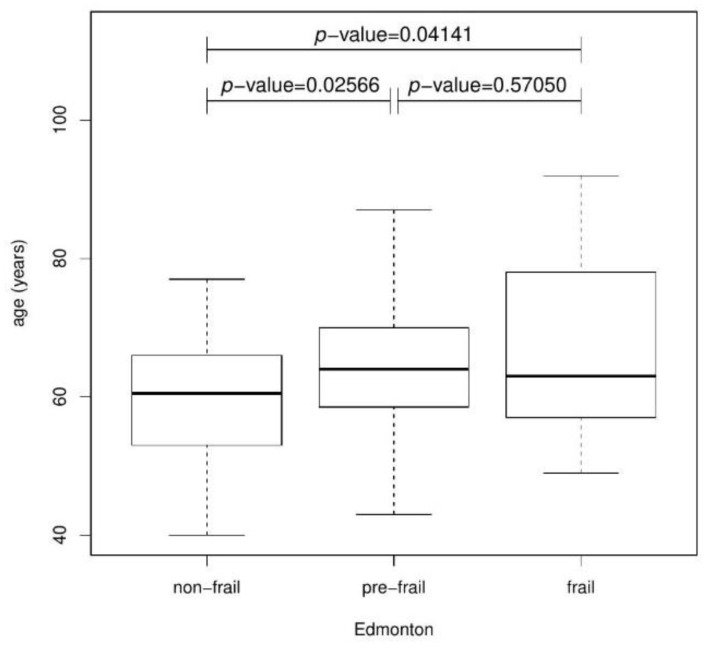
Boxplot presenting the relationship between Age and Edmonton.

**Figure 5 jcm-11-04022-f005:**
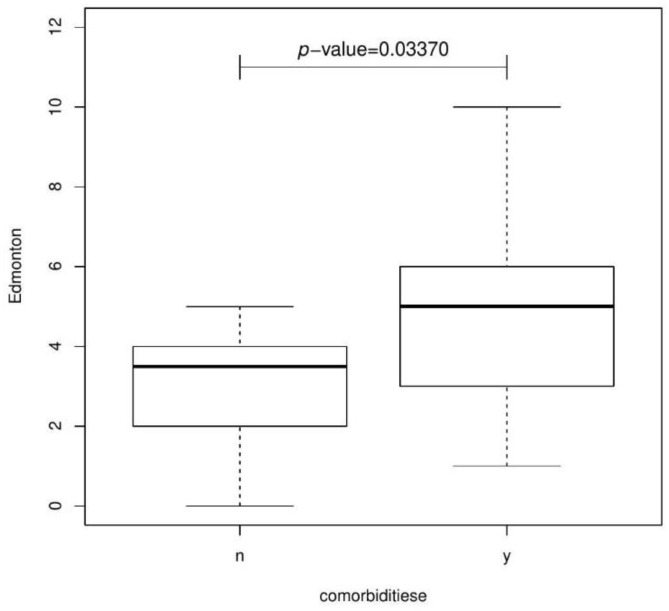
Boxplot presenting the relationship between Comorbidities and Edmonton.

**Figure 6 jcm-11-04022-f006:**
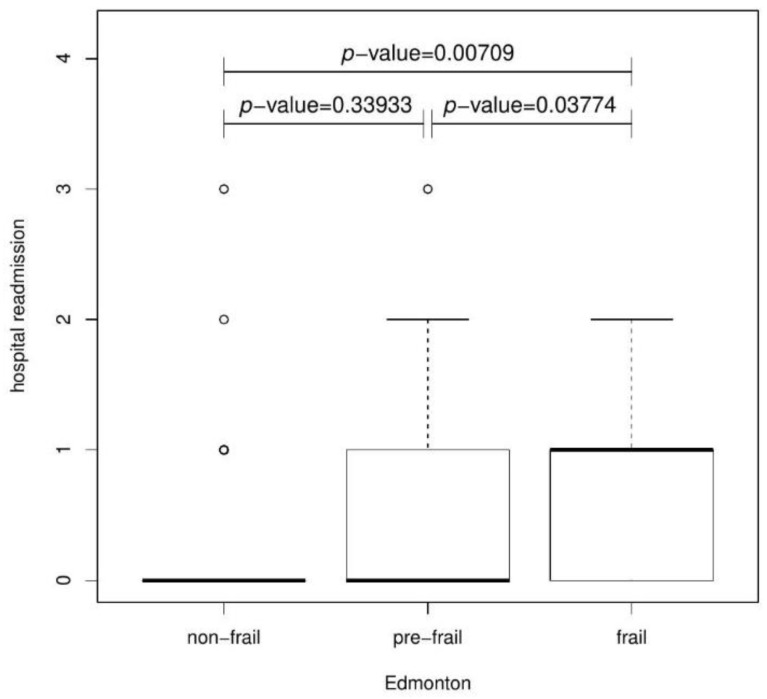
Boxplot presenting the relationship *p* between Hospital readmission and Edmonton after 6 months.

**Figure 7 jcm-11-04022-f007:**
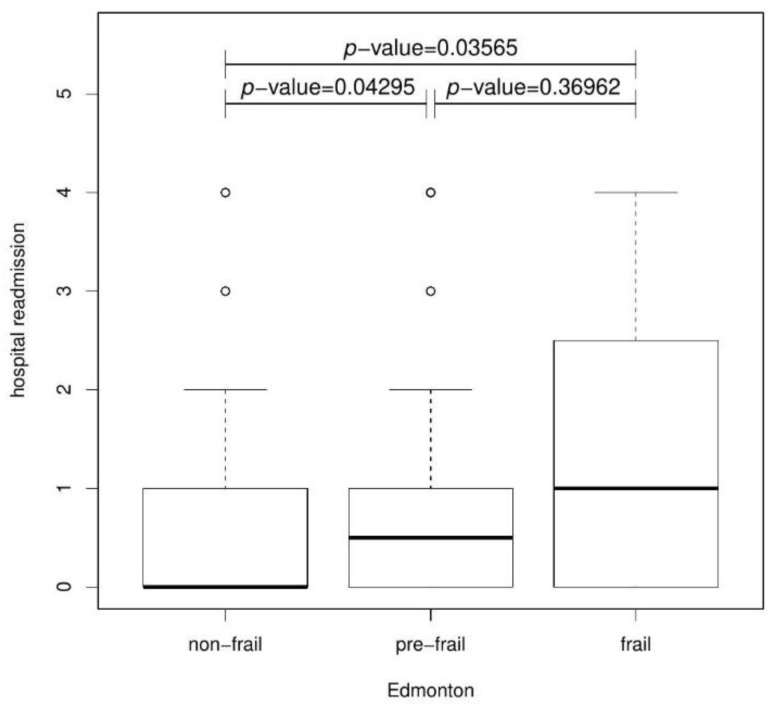
Boxplot presenting the relationship between Hospital readmission and Edmonton after 12 months.

**Table 1 jcm-11-04022-t001:** Characteristics of the study group.

Quantitative Variables	All Patients (*n* = 106) Mean ± Standard Deviation or Median (First Quartile; Third Quartile)
Age (years)	62.6 ± 9.7
Edmonton	5 (3; 6)
Qualitative variables	All patients (*n* = 106) *n* (%)
Cause of hospitalization	
planned	53 (50%)
worsening of heart failure	53 (50%)
Etiology of heart failure	
coronary	70 (66%)
non-coronary	36 (34%)
Comorbidities	
yes	100 (94.3%)
no	6 (5.7%)
Diabetes	
yes	43 (40.6%)
no	63 (59.4%)
Hypertension	
yes	78 (73.6%)
no	28 (26.4%)
Chronic kidney disease	
yes	30 (28.3%)
no	76 (71.7%)
Hypothyroidism	
yes	12 (11.3%)
no	94 (88.7%)
Hyperthyroidism	
yes	3 (2.8%)
no	103 (97.2%)
Stroke	
yes	13 (12.3%)
no	93 (87.7%)
Chronic obstructive pulmonary disease	
yes	17 (16%)
no	89 (84%)
Lower limb atherosclerosis	
yes	15 (14.2%)
no	91 (85.8%)
Implantable cardioverter-defibrillator (ICD)	
yes	52 (49%)
no	54 (51%)
Cardiac resynchronization therapy with defibrillator function (CRT-D)	
yes	17(16%)
no	89 (84%)
NYHA_b- beginning of hospitalization	
II	42 (39.6%)
II/III	5 (4.7%)
III	47 (44.3%)
III/IV	12 (11.3%)
NYHA_e- end of hospitalization	
II	58 (55.8%)
II/III	28 (26.9%)
III	18 (17.3%)
Complications hospitalization	
no	83 (78.3%)
yes	23 (21.7%)

**Table 2 jcm-11-04022-t002:** Edmonton variables contingency table—NYHA class at the beginning of hospitalization.

NYHA_b	Edmonton			*p*-Value Fisher’s Exact Test for Count Data
	Non-Frail	Pre-Frail	Frail	
II	24	17	1	0.00037
II/III	3	2	0	
III	11	27	9	
III/IV	8	1	3	

**Table 3 jcm-11-04022-t003:** Edmonton variables contingency table—complications during hospitalization.

Complications	Edmonton			*p*-Value Fisher’s Exact Test for Count Data
	Non-Frail	Pre-Frail	Frail	
No	40	36	7	0.03875
Yes	6	11	6	

**Table 4 jcm-11-04022-t004:** Contingency table of Edmonton scale variables—NYHA class at discharge.

NYHA_e	Edmonton			*p*-Value Fisher’s Exact Test for Count Data
	Non-Frail	Pre-Frail	Frail	
II	39	19	0	<0.00001
II/III	6	19	3	
III	1	8	9	

**Table 5 jcm-11-04022-t005:** Predicted probabilities derived from the ordinal regression model.

	Probability		
	Non-Frail	Pre-Frail	Frail
Chronic kidney disease yes	0.5482	0.3953	0.0565
No	0.8896	0.1015	0.0089

## Data Availability

The data will be available by contacting the corresponding author.

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
