# Peer review of "Employing the Multivariate Edmonton Scale in the Assessment of Frailty Syndrome in Heart Failure"

_jcm, 2022, doi:10.3390/jcm11144022_

Round 1
Reviewer 1 Report
The Authors have not replied to the 5/6 issues identified in the previous revision. Moreover, the issue of novelty and the additive value and information of the current study have not been adequately addressed.
Author Response
Reviewer 1:
We have collected all 6/6 responses with explanations. In addition, we have completed
some points.
In this study, Studzińska et al. set out to investigate the occurrence of frailty syndrome in
heart failure using the Edmonton Frailty Scale. Despite this study being of some interest there
are several issues that should be addressed.
The main issue of this study is the novelty of the results.
The Authors should further expand on the importance of their findings. In the current state,
their results appear only as confirmatory of previous studies, which have reported that frailty
is associated with worse NYHA status, outcomes during complications and presence of
comorbidities. 1-3
Response: We are aware of the great interest in the topic of frailty syndrome among patients
with heart failure. In the original version of the discussion, we described the existing
difficulty and limited possibilities for comparing intercenter results. The main difficulty is the
variety of assessment tools and the very heterogeneous study groups. In our discussion we
cited the work which does not show correlation between NYHA clasification and the frailty
syndrome[17]- ours demonstrates it. Additionally, experts emphasize the need to assess the
frailty in all patients with failure [2].
By analyzing the occurrence of chronic diseases in patients with heart failure, the results were
indeed similar to those in many studies - also those concerning other groups of subjects
(without heart failure). It should be noted, however, that the weakness syndrome may also
apply to people who have not been with any chronic disease[3]. Based on survey data by
Longitudinal Aging Study Amsterdam (LASA) , it has been revealed that community-based
elderly people with heart failure have an increased risk of weakness, irrespective of other
factors such as gender, age and multi-morbidity ( the median follow-up of the study was 8,4
years)[29].
The topic of complications during hospitalization in patients with heart failure is certainly an
important topic and must be developed and analyzed. There is a need for ongoing education
and awareness of the team caring for the patient (doctors, nurses or physiotherapists)
regarding the specific individual capabilities and needs of patients with heart failure.
The obtained results concerning the course of hospitalization and rehospitalization are
consistent with the results of other authors. On this basis, however, we may conclude that the
Edmonton scale is a useful, simple tool for applaying in the hospital setting. Due to its
multidimensional nature, the information obtained from the study can be used by members of
the medical team to determine and organize the care and treatment of patients with heart
failure during hospitalization. It can also provide valuable information about the patient to
physicians, nurses, physiotherapists and psychologists. Due to the fact that one specific scale
for the assessment of frailty in heart failure has not been confirmed, the assessment of frailty
among patients hospitalized with heart failure using the Edmonton scale seems to be a
valuable and necessary issue. So far, we managed to find one publication using the Edmonton
scale among elderly patients >= 80 years in the cardiological intensive care unit. (DOI:
10.1016/j.cjco.2021.03.009). Different scoring thresholds for the scale and different age
groups make it difficult to compare our studies.
Other issues:
1. The prevalence of frail patients is relatively low (12.3%) compared to other cohorts
describing frailty in HF. 1 How would you explain this difference?
Response: The fact that frailty in the research group is less common could result from the
selection of the research group. The average age of patients was 62.6 ± 9.7 years. In
publications the fact of the influence of age on frailty syndrom is often stressed and frequency
of diagnosis increases with age. In addition, the fact that the study involved a relatively small
group of patients with reduced ejection fraction only may have had an impact. In the available
studies the prevalence of frailty in this group of HFrEF patients is lower than in HFpEF.[7,
https://doi.org/10.1002/ehf2.13885]. This relationship is also explained by the more advanced
age of patients with HFpEF.
2.Have you implemented any statistical power calculations for his study? For example,
you mention that complications occurred more frequently in frail patients. However, the
actual number with frailty and HF is only six patients.
Response: The ‘Mkpower’ package and the sim.power.wilcox.test procedure were used to
calculate the power of the Wilcoxon test. The power of the test for Edmonton values divided
into groups according to the values of the dichotomous variable 'comorbidities' is equal to
0.5928. The limitations and assumptions related to the use of the sim.power.wilcox.test
procedure mean that the obtained result should be interpreted taking into account all these
limitations.
3. How exactly were complications during the hospital stay defined and monitored?
Response: Each patient during hospitalization has a detailed medical documentation.
Additionally, the authors of the publication conducted an interview with all study participants,
in which they were asked about the course of hospitalization. The incidence of infection,
bleeding and trauma as well as all other complications combined were
monitored. Complications were recorded by interview and analysis of medical records during
the patient's hospital stay.
4. Why were both patients admitted to the hospital on a scheduled basis and patients with
exacerbation of HF both included? Please describe the reasons for admission for the
scheduled patients.
Response: The research was carried out at the Cardiology Clinic of a multi-specialist clinical
hospital at the Medical University of Gdańsk. The group of patients scheduled to come to the
hospital were people under 65 years of age, whose hospitalization was related to the
qualification for heart transplantation. Thus we were able to examine people of different age
with a low ejection fraction, which was the assumption of the project. The study for all the
patients took place at the end of hospitalization whereas patients admitted with an
exacerbation of the disease were examined after their stabilization.
5. Why were patients with EF>40% excluded? Would the inclusion of EF>40% increase
the study’s population? How was information about EF obtained?
Response: The publication was supplemented .
Limiting the study group to patients with EF 40% was purposeful. Information on the
ejection fraction was obtained from the echocardiography. The study was conducted during
patients' hospitalization, in patients with disease exacerbation at the end of hospitalization and
after clinical improvement. The inclusion of subjects with EF>40% would increase the study
population but would also couse the heterogeneity of the group. Since, according to the
available knowledge, the occurrence of weakness can be influenced by EF, we wanted to
study a selected group of patients.
6. The statistical analysis of the study is overall of poor quality. The Authors should
provide at least a regression analysis aiming to identify the determinants of frailty
based on comorbidities.
Response: The publication was supplemented material with the above-mentioned suggestion.
The ordinal regression model was used to determine the relationship between the groups of
the Edmonton variable and dichotomous variables describing the existence of comorbidities
of the studied patients. Eight models were created in which the dependent variable were
groups of the Edmonton variable and the independent variable was one of the eight
comorbidities listed in Table 1. The coefficients of the model are statistically significant only
for the variable 'chronic kidney disease' (p-value = 0.00004). The odds ratio OR = 6.6423 was
determined using the value of the coefficient for the variable 'chronic kidney disease'
(1.8935). The obtained result means that the odds that a person suffering from chronic kidney
disease will be in the group with higher values of the Edmonton variable is over 6.6 times
higher than for patients who do not suffer from the above-mentioned disease. The predicted
probability of belonging of patients suffering from chronic
kidney disease to particular groups of the Edmonton variable is presented in Table.
Predicted probabilities derived from the ordinal regression model.
probability
non-frail pre-frail frail
Chronic kidney disease yes
no
0.5482 0.3953 0.0565
0.8896 0.1015 0.0089
A multivariate model was also developed with all comorbidities studied as independent
variables. In this model, the determined coefficients were statistically significant (p-value =
0.00002 and p-value = 0.0412, respectively) for the variables 'chronic kidney disease' and
'hypertension'. The odds that a person suffering from chronic kidney disease will be in the
group with higher values of the Edmonton variable is almost 8 times, and in the case of
hypertension, 3 times higher than for patients who do not suffer from the above mentioned
diseases.
The spelling was checked throughout the article.

Reviewer 2 Report
The manuscript entitled 'Employing the multivariate Edmonton scale in the assessment of the frailty syndrome in heart failure' studied the association of heart failure complexities with Frailty. The study utilizes 106 patients in the analysis. A multivariate Edmonton scale has been used to assess the Frailty. The frailty data was compared with the occurrence of complications. Though the manuscript provides various types of analysis, there are a few concerns that should be addressed:
1. The objective of the manuscript is not clear. The beginning of the manuscript is 'Frailty syndrome has a significant impact on the prognosis in patients with 14 heart failure.' However, the study focused on analyzing the Frailty's progression in heart failure patients. Therefore, it is critical to define if heart failure exacerbates the Frailty. If yes, then the manuscript should be accordingly modified.
2. More than 94% of patients have a comorbidity of other diseases. Moreover, most of them are age-associated and already known risk factors for decreased cardiac function and heart failure (HF). Therefore, the inclusion of the no frail control for with no heart failure is essential to conclude if frail increases the HF risk.
3. Define the Frailty and elaborate on the specific reasons why Frailty should be analyzed in HF patients or vice versa.
4. Please elaborate on how this study is different than the previous similar studies.
5. Provide data showing the association of increased Frailty with reduced ejection fraction.
6. The study analyzed various parameters of Frailty. Is there any specific parameter that has a greater association with the complexity than others?
7. What is NYHA?
Author Response
Rewiever 2:
The manuscript entitled 'Employing the multivariate Edmonton scale in the assessment of the
frailty syndrome in heart failure' studied the association of heart failure complexities with
Frailty. The study utilizes 106 patients in the analysis. A multivariate Edmonton scale has
been used to assess the Frailty. The frailty data was compared with the occurrence of
complications. Though the manuscript provides various types of analysis, there are a few
concerns that should be addressed:
1. The objective of the manuscript is not clear. The beginning of the manuscript is 'Frailty
syndrome has a significant impact on the prognosis in patients with 14 heart failure.'
However, the study focused on analyzing the Frailty's progression in heart failure patients.
Therefore, it is critical to define if heart failure exacerbates the Frailty. If yes, then the
manuscript should be accordingly modified.
Response: The Abstract was modified.
The aim of the introduction was to show the complexity of frailty in patients with heart
failure. This article did not focus on the analysis of the progression of frailty, but only on the
usefulness of the Edmonton Frailty Scale to assess hospitalised patients. In order to address
the comment, the conclusion was modified and the phrase prognosis was removed, as indeed
this term does not refer to our results. Actually, the aim of the study was not to assess the
prognosis of patients but the usefulness of the EFS scale for use during hospitalization. The
results of rehospitalization were intentionally included (fig 6, fig7).These data show the
monitoring of the patient's condition and not the prognosis. Regarding the prognosis, we
relied on the available literature. We hope that the explanation is now sufficiently precise.
2. More than 94% of patients have a comorbidity of other diseases. Moreover, most of them are
age-associated and already known risk factors for decreased cardiac function and heart failure
(HF). Therefore, the inclusion of the no frail control for with no heart failure is essential to
conclude if frail increases the HF risk.
Response: In our study and results we did not address the assessment of increasing the risk of
frailty in HF. We quote such a sentence in the discussion referring to available publications
and studies in this area [7].
We are aware of the high prevalence of comorbidities in patients with HF. Although it is the
group of patients with HFpEF that is identified as the one more burdened by comorbidities.
Our study focused on HFrEF. Furthermore, the papers highlight the fact that frailty can affect
people with heart failure regardless of factors such as age or multimorbidity defined as having
at least two chronic diseases [35].
Another study concluded that although frailty and multimorbidity are overlapping and
complementary concepts, they remain distinct entities because patients with multimorbidity
may or may not be frail and vice versa[ https://doi.org/10.1093/gerona/glx178, doi:
10.1093/gerona/56,3.m146.
3. Define the Frailty and elaborate on the specific reasons why Frailty should be analyzed in
HF patients or vice versa.
Response: In the introduction of the paper we presented the concept of frailty. We considered
two concepts - frailty in the context of physical limitations and frailty in a multidimensional
context. In addition, we cited the most up-to-date definition described in the paper by the
Heart Failure Society of the E S C. The published document they define frailty as “a
multidimensional dynamic state, independent of age, which makes a patient with heart failure
susceptible to stressors” [7]. The occurrence of a factor (stressor), e.g. infection, drug regime
change, is associated with a disproportionate reaction, excessive deterioration of health, and
thus increases the risk of decompensation. Complications of the weakness include: reduced
fitness and independence, greater risk of falls, fractures, greater susceptibility to disease and
more frequent hospitalizations, as well as the need for institutional care and premature death
[8].
The need to take into account frailty in patients is included in the current guidelines for the
treatment of heart failure. These include the European Society of Cardiology guidelines
(2021), A Report of the American College of Cardiology/American Heart Association Joint
Committee on Clinical Practice Guidelines (2022), Comprehensive Update of the Canadian
Cardiovascular Society Guidelines for the Management of Heart Failure (2017). In addition,
the Canadian Cardiovascular Society in the same document recommends the use of the
Edmonton Frailty Scale to assess frailty.
Assessment of frailty also appears to be important because of the inverse relationship that
exists. Individuals with frailty syndrome and even patients who belong to the pre-frail group
and exhibit features of vulnerability have a significantly increased risk of developing
cardiovascular disease including heart failure.
4.Please elaborate on how this study is different than the previous similar studies.
Response: In the presented reviews and meta-analyses of many studies on patients with HF, in
most of them the assessment of frailty was based on the analysis of physical limitations [22,
https://doi.org/10.1155/2018/8739058]. Our article, as recommended by the Heart Failure
Association, focuses on a more accurate multidimensional concept and addresses the different
dimensions of frailty (physical, psychological , social). The available studies also cover very
diverse study groups, some focused on patients with HF with preserved ejection fraction
(HFpEF), others including patients without LVEF measurement. In addition, the difference is
based on the study site.It concerned less the patients who present to hospital with acute
decompensation than the well-aligned outpatients [DOI: 10.15420/cfr.2021.29]
There is currently no recognised gold standard for assessing frailty and no frailty assessment
tool has been specifically validated in the HF population[ 7,12]. The researchers emphasise
that assessment of frailty in HF patients needs to be standardised and unified. Otherwise,
differences caused by measurement methods may not fully reflect the patient's condition
[https://doi.org/10.1155/2018/8739058 ]. Edmonton, appears to be a simple tool, allowing
assessment in a hospital setting (as in our patient group). S. Pern's study suggests that
measuring frailty with the EFS is a helpful tool for assessing frailty in a group of
institutionalised older people [ doi: 10.1186/s12877-016-0382-3.]. In addition, the EFS score
correlates with the results of other tests such as Mini-Mental State Examination(MMSE),
functional independence (ADL, IADL scales), medication use (counting medications taken
daily), Mini Nutrition Assessment (MNA), Geriatric Depression Scale (GDS). So far, we
managed to find one publication using the Edmonton scale among elderly patients >= 80
years in the cardiological intensive care unit. (DOI: 10.1016/j.cjco.2021.03.009). Different
scoring thresholds for the scale and different age groups make it difficult to compare our
studies.
5. Provide data showing the association of increased Frailty with reduced ejection fraction.
Response: An important position statement from the Heart Failure Association (HFA) of the
ESC cited earlier stated that the weakness appears to be more common in HF patients with
preserved ejection fraction (HFpEF) than in those with reduced ejection fraction (HFrEF).
The authors associated this relationship with a higher burden of comorbidities and older age
of patients with HFpEF [7].The same findings were presented in subsequent publications [ 37,
DOI: 10.15420/cfr.2021.29, https://doi.org/10.1002/ehf2.13885 ]. In another source among
hospitalised HF patients, the burden of frailty is similar in HFpEF compared with HFrEF [37].
We report several studies relating to the assessment of frailty in patients with reduced EF.
According to Kang et al., the presence of frailty may be associated with a poor prognosis during
hospitalization of over 60 years old patients with heart failure and a reduced ejection fraction
[31].
Analysis of the PARADIGM-HF and ATMOSPHERE studies included 13265 patients with
HFrEF [36]. The results are as follows: 63% of patients in this group were frail. Frailty is
associated with greater deterioration in quality of life and a higher risk of hospitalisation and
death. Strategies to prevent and treat frailty are needed in HFrEF.
Results of studies by other authors: Patients with HFrEF with a high burden of frailty have a
significantly higher risk for adverse clinical outcomes and are less likely to be initiated and
up-titrated on an optimal guideline-directed medical therapy (GDMT) regimen.
DOI:10.1016/j.jchf.2021.12.004
Two of these papers were cited in our article.
6. The study analyzed various parameters of Frailty. Is there any specific parameter that has a
greater association with the complexity than others?
Response: The values of the parameter η2
(effect size for the Kruskal-Wallis test) were
determined for the variables "age" (η2 = 0.0472), "hospital readmission
after 6 months" (η2 = 0.0580), "hospital readmission after 12 months" (η2 =
0.0321). In each case, the obtained values of η2 belong to the "small"
category.
In the case of Fisher's Exact Test for Count Data, it is only possible to
analyze the obtained association plots. On their basis, it can be concluded
that the presented residuals for the "NYHA_e" variable are larger than for
the other analyzed variables.
7. What is NYHA?
Response: An explanation has been included in the paper. All abbreviations used in the paper are
explained.
The spelling was checked throughout the article

Round 2
Reviewer 1 Report
The Authors have adequately answered all my comments
Author Response
Thank you very much for your positive review.
This manuscript is a resubmission of an earlier submission. The following is a list of the peer review reports and author responses from that submission.
Round 1
Reviewer 1 Report
Thank you for allowing me to review this very interesting article. Studzinska and colleagues have employed the Edmonton Frailty Scale in frailty assessment for patients with heart failure. They studied 106 patients with LVEF < 40%, and found a correlation between NYHA class and frailty. They also saw that patients higher up on the frailty scale were more prone to developing complications while they were hospitalized. This is an important topic, and I congratulate the authors for providing research on frailty, something that is not as readily discussed as it should be, when talking about patients with heart failure.
While I do find this very intriguing, I have a few questions and concerns for the authors.
Major
- The Edmonton Frailty Scale is a subjective questionnaire which takes into account several factors such as functional independence, nutrition, social support, mood etc. NYHA classification is fairly objective in nature. The EFS could change depending upon how the patient is feeling, what is his current social status etc, but the NYHA will not. How would you explain the correlation between something so subjective, and something objective?
- If we are to say that the EFS can predict which patients will have complications during the hospital stay or have a bad outcome, how will it affect clinical management?
- What is the significance of measuring the correlation at the end of hospitalization?
- Did any of these patients undergo surgery for heart failure? If yes, were outcomes different based on EFS?
Minor - Introduction and Discussion, both need to be curtailed.
Author Response
Thank you for your review and some valuable tips, according to which we could improve our work.
An 1,3 The research involved patients admitted to a hospital as planned (qualified for a heart transplantation) and patients with worsening of heart failure. The two of NYHA assessments were included in the study. One at the admission to a hospital and the second just before leaving the hospital ward (after stabilizing the patients with worsening of the heart failure). The frailty assessment (Edmonton scale) was carried out deliberately just before leaving the hospital. That’s because we wanted our assessment to be the most objective.
An 2 According to the current knowledge patients with frailty syndrome are patients who have troubles with adherence to treatment recommendations. In case of deterioration , they show greater risk of long-term hospitalizations or even premature death. Frailty may cause the loss of independence, predisposes to falls and fractures. It increases the need of long-term care and it has negative impact on the quality of life. Diagnosis of frailty or identification of pre-frail patients (using multidimensional EFS scale) allows the patients to be provided with intensive, directional care of the whole medical team including nurses, physiotherapists, dieticians and psychologists. This will allow to identify individual problems and needs of patients ,as well as plan further optimal therapy.
An4 Patients who undergone an operation during their hospital stay , were not included in this study.
An 5 There has some difficulty appeared. One of the reviewers asked for extending an introduction and discussion, while you are asking, sir, for the shortened version. Such a situation makes it impossible to adhere to comments of the two reviewers. We are asking you to agree on leaving the introduction and discussion in the previous form. The cause is our lack of possibility to adhere to opposite remarks.
Reviewer 2 Report
In this study, Studzińska et al. set out to investigate the occurrence of frailty syndrome in heart failure using the Edmonton Frailty Scale. Despite this study being of some interest there are several issues that should be addressed.
The main issue of this study is the novelty of the results.
The Authors should further expand on the importance of their findings. In the current state, their results appear only as confirmatory of previous studies, which have reported that frailty is associated with worse NYHA status, outcomes during complications and presence of comorbidities. 1-3
Other issues:
- The prevalence of frail patients is relatively low (12.3%) compared to other cohorts describing frailty in HF. 1 How would you explain this difference?
- Have you implemented any statistical power calculations for his study? For example, you mention that complications occurred more frequently in frail patients. However, the actual number with frailty and HF is only six patients.
- How exactly were complications during the hospital stay defined and monitored?
- Why were both patients admitted to the hospital on a scheduled basis and patients with exacerbation of HF both included? Please describe the reasons for admission for the scheduled patients.
- Why were patients with EF>40% excluded? Would the inclusion of EF>40% increase the study’s population? How was information about EF obtained?
- The statistical analysis of the study is overall of poor quality. The Authors should provide at least a regression analysis aiming to identify the determinants of frailty based on comorbidities.
Author Response
Thank you very much for your review and some valuable tips, according to which we could improve our work. Below we will respond to specific comments and concerns that have been pointed out, thanks to which, in our opinion, the work has gained a lot.
An1. The fact that frailty in the research group is less common could result in the selection of the research group. The average age of patients was 62.6 ± 9.7 years. In publications the fact of the influence of age on frailty syndrom is often stressed and frequency of diagnosis increases with age.
An2.
The ‘Mkpower’ package and the sim.power.wilcox.test procedure were used to calculate the power of the Wilcoxon test. The power of the test for Edmonton values divided into groups according to the values of the dichotomous variable 'comorbidities' is equal to 0.5928. The limitations and assumptions related to the use of the sim.power.wilcox.test procedure mean that the obtained result should be interpreted taking into account all these limitations.
An. 3
Each patient during hospitalization has a detailed medical documentation. Additionally, the authors of the publication conducted an interview with all study participants, in which they were asked about the course of hospitalization. The incidence of infection, bleeding and trauma as well as all other complications combined were monitored. Complications were recorded by interview and analysis of medical records during the patient's hospital stay.
An. 4 The research was carried out at the Cardiology Clinic of a multi-specialist clinical hospital at the Medical University of Gdańsk. The group of patients scheduled to come to the hospital were people under 65 years of age, whose hospitalization was related to the qualification for heart transplantation. Thanks to this, we were able to examine people of different age (with a low ejection fraction), which was the assumption of the project. The study for all patients took place at the end of hospitalization and after stabilization of patients admitted with an exacerbation of the disease.
An. 5 Limiting the study group to patients with EF 40% was purposeful. Information on the ejection fraction was obtained from the echocardiography. The inclusion of subjects with EF>40% would increase the study population but would also increase the heterogeneity of the group.
An6. The publication was supplemented material with the above-mentioned suggestion.
The ordinal regression model was used to determine the relationship between the groups of the Edmonton variable and dichotomous variables describing the existence of comorbidities of the studied patients. Eight models were created in which the dependent variable were groups of the Edmonton variable and the independent variable was one of the eight comorbidities listed in Table 1. The coefficients of the model are statistically significant only for the variable 'chronic kidney disease' (p-value = 0.00004). The odds ratio OR = 6.6423 was determined using the value of the coefficient for the variable 'chronic kidney disease' (1.8935). The obtained result means that the odds that a person suffering from chronic kidney disease will be in the group with higher values of the Edmonton variable is over 6.6 times higher than for patients who do not suffer from the above-mentioned disease. The predicted probability of belonging of patients suffering from chronic
kidney disease to particular groups of the Edmonton variable is presented in Table.
Predicted probabilities derived from the ordinal regression model.
|
|
probability |
||
|
non-frail |
pre-frail |
frail |
|
|
Chronic kidney disease yes no |
0.5482 |
0.3953 |
0.0565 |
|
0.8896 |
0.1015 |
0.0089 |
|
A multivariate model was also developed with all comorbidities studied as independent variables. In this model, the determined coefficients were statistically significant (p-value = 0.00002 and p-value = 0.0412, respectively) for the variables 'chronic kidney disease' and 'hypertension'. The odds that a person suffering from chronic kidney disease will be in the group with higher values of the Edmonton variable is almost 8 times, and in the case of hypertension, 3 times higher than for patients who do not suffer from the above mentioned diseases.
Reviewer 3 Report
Studzińska et al. present an interesting analysis of frailty syndrome among patients with HFrEF. The study design is well. The paper is well written.
I have the following suggestions
1-Abstract: please add some baseline characterisics e.g. age, gender, etiology of heart failure
Do you have any outcome data? e.g. rehospitalization for cardiovascular event?
2-Introduction:
of note frailty could be triggered by lost of muscule in HFrEF.
What about drugs e.g. sacubitril/valsartan? data have shown the beneficial effect of such drug discuss these papers (PMID: 32186406, PMID: 34768510 ).
Result and discussion
Is frailty more presented at higher age? Of note, some groups described age differences regarding to the use of heart failure drugs
HFrEF is associated with ventricular tachyarrhythmias. Any data on in your cohort about how many patients received cardiac electronic device, CRTP, CRTD?
how many patients suffered from ventricular tachyarrhythmias?
Which heart failure drugs were presecribed?
Data have shown that e.g. sacubitril/valsartan may impact the risk of ventricular tachyarrhythmias. This should also be discussed.
Please disucss how frailty could impact the management of HFrEF.
The limitation section should be modified.
Author Response
Thank you very much for your review and valuable tips, according to which we could improve our work. Below we will respond to specific comments and concerns that have been pointed out, thanks to which, in our opinion, the work has gained a lot.
An. 1
Abstract- the publication was completed with the above-mentioned suggestion.
An2
This is obviously a very valuable observation. We have follow-up data from the study group at 6 and 12 months. The aim of the current study was to assess the occurrence of the frailty syndrome in heart failure using the multidimensional Edmonton Frailty Scale. There are further publications using rehospitalisation data planned.
An 3
Thank you for your suggestion. The authors alluded to a very important point in their introduction - the possibility of overlapping symptoms including reduced muscle mass in heart failure and frailty.
There has some difficulty appeared. One of the reviewers asked for shortening an introduction and discussion, while you, sir, are asking for the extended version. Such a situation makes it impossible to adhere to comments of the two reviewers. We are asking you to agree on leaving the introduction and discussion in the previous form. The cause is our lack of possibility to adhere to opposite remarks.
An 4, 9
We are aware of high efficiency of drugs such as sacubitril/valsartan. The reserch was carried out between 2016 and 2020 when the drugs were not avaible due to its high cost.
An 5, 10
The discussion focused on the results obtained in relation to the literature. Evaluation of pharmacotherapy was not the aim of this study. It wasn't our purpose to asses the improvement or deterioration of the patient. We have just assessed the occurrence of the frailty syndrome in heart failure using the multidimensional Edmonton Frailty Scale.
There has some difficulty appeared. One of the reviewers asked for shortening an introduction and discussion, while you, sir, are asking for the extended version. Such a situation makes it impossible to adhere to comments of the two reviewers. We are asking you to agree on leaving the introduction and discussion in the previous form. The cause is our lack of possibility to adhere to opposite remarks.
An 6 The publication was completed with the above-mentioned suggestion.
An 7 We haven’t got any information on how many patients suffered from ventricular tachyarrhythmia. The work was completed with data on number of patients with ICD and CRT-D.
An 8
Patients were treated according to the then current ECS guidelines for the diagnosis and treatment of acute and chronic heart failure (2016).
An 11
This study has a limitations that need to be highlighted. Most important of which was the very low size of the research group. The more participants the better value and quality of the research.
Round 2
Reviewer 1 Report
The authors have satisfactorily addressed my concerns.